# Conic Scan-and-Cover algorithms for nonparametric topic modeling

**Mikhail Yurochkin**
Department of Statistics
University of Michigan
moonfolk@umich.edu

**Aritra Guha**
Department of Statistics
University of Michigan
aritra@umich.edu

**XuanLong Nguyen**
Department of Statistics
University of Michigan
xuanlong@umich.edu

## Abstract

We propose new algorithms for topic modeling when the number of topics is unknown. Our approach relies on an analysis of the concentration of mass and angular geometry of the topic simplex, a convex polytope constructed by taking the convex hull of vertices representing the latent topics. Our algorithms are shown in practice to have accuracy comparable to a Gibbs sampler in terms of topic estimation, which requires the number of topics be given. Moreover, they are one of the fastest among several state of the art parametric techniques.[1] Statistical consistency of our estimator is established under some conditions.

## 1 Introduction

A well-known challenge associated with topic modeling inference can be succinctly summed up by the statement that sampling based approaches may be accurate but computationally very slow, e.g., Pritchard et al. (2000); Griffiths & Steyvers (2004), while the variational inference approaches are faster but their estimates may be inaccurate, e.g., Blei et al. (2003); Hoffman et al. (2013). For nonparametric topic inference, i.e., when the number of topics is a priori unknown, the problem becomes more acute. The Hierarchical Dirichlet Process model (Teh et al., 2006) is an elegant Bayesian nonparametric approach which allows for the number of topics to grow with data size, but its sampling based inference is much more inefficient compared to the parametric counterpart. As pointed out by Yurochkin & Nguyen (2016), the root of the inefficiency can be traced to the need for approximating the posterior distributions of the latent variables representing the topic labels — these are not geometrically intrinsic as any permutation of the labels yields the same likelihood.

A promising approach in addressing the aforementioned challenges is to take a *convex geometric* perspective, where topic learning and inference may be formulated as a convex geometric problem: the observed documents correspond to points randomly drawn from a *topic polytope*, a convex set whose vertices represent the topics to be inferred. This perspective has been adopted to establish posterior contraction behavior of the topic polytope in both theory and practice (Nguyen, 2015; Tang et al., 2014). A method for topic estimation that exploits convex geometry, the Geometric Dirichlet Means (GDM) algorithm, was proposed by Yurochkin & Nguyen (2016), which demonstrates attractive behaviors both in terms of running time and estimation accuracy. In this paper we shall continue to amplify this viewpoint to address *nonparametric topic modeling*, a setting in which the number of topics is unknown, as is the distribution inside the topic polytope (in some situations).

We will propose algorithms for topic estimation by explicitly accounting for the concentration of mass and angular geometry of the topic polytope, typically a simplex in topic modeling applications. The geometric intuition is fairly clear: each vertex of the topic simplex can be identified by a ray emanating from its center (to be defined formally), while the concentration of mass can be quantified

for the cones hinging on the apex positioned at the center. Such cones can be rotated around the center to scan for high density regions inside the topic simplex — under mild conditions such cones can be constructed efficiently to recover both the number of vertices and their estimates.

We also mention another fruitful approach, which casts topic estimation as a matrix factorization problem (Deerwester et al., 1990; Xu et al., 2003; Anandkumar et al., 2012; Arora et al., 2012). A notable recent algorithm coming from the matrix factorization perspective is RecoverKL (Arora et al., 2012), which solves non-negative matrix factorization (NMF) efficiently under assumptions on the existence of so-called anchor words. RecoverKL remains to be a parametric technique — we will extend it to a nonparametric setting and show that the anchor word assumption appears to limit the number of topics one can efficiently learn.

Our paper is organized as follows. In Section 2 we discuss recent developments in geometric topic modeling and introduce our approach; Sections 3 and 4 deliver the contributions outlined above; Section 5 demonstrates experimental performance; we conclude with a discussion in Section 6.

## 2 Geometric topic modeling

**Background and related work** In this section we present the convex geometry of the Latent Dirichlet Allocation (LDA) model of Blei et al. (2003), along with related theoretical and algorithmic results that motivate our work. Let $V$ be vocabulary size and $\Delta^{V-1}$ be the corresponding vocabulary probability simplex. Sample $K$ topics (i.e., distributions on words) $\beta_k \sim \text{Dir}_V(\eta)$, $k = 1, \ldots, K$, where $\eta \in \mathbb{R}_+^V$. Next, sample $M$ document-word probabilities $p_m$ residing in the *topic simplex* $B := \text{Conv}(\beta_1, \ldots, \beta_K)$ (cf. Nguyen (2015)), by first generating their *barycentric coordinates* (i.e., topic proportions) $\theta_m \sim \text{Dir}_K(\alpha)$ and then setting $p_m := \sum_k \beta_k \theta_{mk}$ for $m = 1, \ldots, M$ and $\alpha \in \mathbb{R}_+^K$. Finally, word counts of the $m$-th document can be sampled $w_m \sim \text{Mult}(p_m, N_m)$, where $N_m \in \mathbb{N}$ is the number of words in document $m$. The above model is equivalent to the LDA when individual words to topic label assignments are marginalized out.

Nguyen (2015) established posterior contraction rates of the topic simplex, provided that $\alpha_k \leq 1 \, \forall k$ and *either* number of topics $K$ is known *or* topics are sufficiently separated in terms of the Euclidean distance. Yurochkin & Nguyen (2016) devised an estimate for $B$, taken to be a fixed unknown quantity, by formulating a geometric objective function, which is minimized when topic simplex $B$ is close to the normalized documents $\bar{w}_m := w_m/N_m$. They showed that the estimation of topic proportions $\theta_m$ given $B$ simply reduces to taking barycentric coordinates of the projection of $\bar{w}_m$ onto $B$. To estimate $B$ given $K$, they proposed a Geometric Dirichlet Means (GDM) algorithm, which operated by performing a k-means clustering on the normalized documents, followed by a geometric correction for the cluster centroids. The resulting algorithm is remarkably fast and accurate, supporting the potential of the geometric approach. The GDM is not applicable when $K$ is unknown, but it provides a motivation which our approach is built on.

**The Conic Scan-and-Cover approach** To enable the inference of $B$ when $K$ is not known, we need to investigate the concentration of mass inside the topic simplex. It suffices to focus on two types of geometric objects: cones and spheres, which provide the basis for a complete coverage of the simplex. To gain intuition of our procedure, which we call Conic Scan-and-Cover (CoSAC) approach, imagine someone standing at a center point of a triangular dark room trying to figure out all corners with a portable flashlight, which can produce a *cone* of light. A room corner can be identified with the direction of the farthest visible data objects. Once a corner is found, one can turn the flashlight to another direction to scan for the next ones. See Fig. 1a, where red denotes the scanned area. To make sure that all corners are detected, the cones of light have to be open to an appropriate range of angles so that enough data objects can be captured and removed from the room. To make sure no false corners are declared, we also need a suitable stopping criterion, by relying only on data points that lie beyond a certain spherical radius, see Fig. 1b. Hence, we need to be able to gauge the concentration of mass for suitable cones and spherical balls in $\Delta^{V-1}$. This is the subject of the next section.

## 3 Geometric estimation of the topic simplex

We start by representing $B$ in terms of its convex and angular geometry. First, $B$ is centered at a point denoted by $C_p$. The centered probability simplex is denoted by $\Delta_0^{V-1} := \{x \in \mathbb{R}^V | x + C_p \in \Delta^{V-1}\}$.

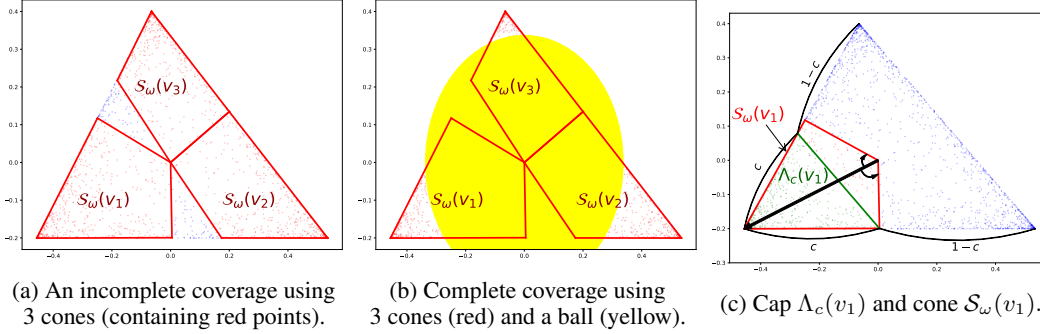

(a) An incomplete coverage using 3 cones (containing red points).

(b) Complete coverage using 3 cones (red) and a ball (yellow).

(c) Cap $\Lambda_c(v_1)$ and cone $\mathcal{S}_\omega(v_1)$.

Figure 1: Complete coverage of topic simplex by cones and a spherical ball for $K = 3$, $V = 3$.

Then, write $b_k := \beta_k - C_p \in \Delta_0^{V-1}$ for $k = 1, \ldots, K$ and $\tilde{p}_m := p_m - C_p \in \Delta_0^{V-1}$ for $m = 1, \ldots, M$. Note that re-centering leaves corresponding barycentric coordinates $\theta_m \in \Delta^{K-1}$ unchanged. Moreover, the extreme points of centered topic simplex $\tilde{B} := \mathrm{Conv}\{b_1, \ldots, b_K\}$ can now be represented by their directions $v_k \in \mathbb{R}^V$ and corresponding radii $R_k \in \mathbb{R}_+$ such that $b_k = R_k v_k$ for any $k = 1, \ldots, K$.

### 3.1 Coverage of the topic simplex

The first step toward formulating a CoSAC approach is to show how $\tilde{B}$ can be *covered* with exactly $K$ cones and one spherical ball positioned at $C_p$. A cone is defined as set $\mathcal{S}_\omega(v) := \{p \in \Delta_0^{V-1} | d_{\cos}(v, p) < \omega\}$, where we employ the angular distance (a.k.a. cosine distance) $d_{\cos}(v, p) := 1 - \cos(v, p)$ and $\cos(v, p)$ is the cosine of angle $\angle(v, p)$ formed by vectors $v$ and $p$.

**The Conical coverage** It is possible to choose $\omega$ so that the topic simplex can be covered with exactly $K$ cones, that is, $\bigcup_{k=1}^{K} \mathcal{S}_\omega(v_k) \supseteq \tilde{B}$. Moreover, each cone contains exactly one vertex. Suppose that $C_p$ is the *incenter* of the topic simplex $\tilde{B}$, with $r$ being the inradius. The incenter and inradius correspond to the maximum volume sphere contained in $\tilde{B}$. Let $a_{i,k}$ denote the distance between the $i$-th and $k$-th vertex of $\tilde{B}$, with $a_{min} \le a_{i,k} \le a_{max}$ for all $i, k$, and $R_{max}, R_{min}$ such that $R_{min} \le R_k := \|b_k\|_2 \le R_{max} \ \forall \ k = 1, \ldots, K$. Then we can establish the following.

**Proposition 1.** For simplex $\tilde{B}$ and $\omega \in (\omega_1, \omega_2)$, where $\omega_1 = 1 - r/R_{max}$ and $\omega_2 = \max\{(a_{min}^2)/(2R_{max}^2), \max_{i,k=1,\ldots,K}(1 - \cos(b_i, b_k))\}$, the cone $\mathcal{S}_\omega(v)$ around any vertex direction $v$ of $\tilde{B}$ contains exactly one vertex. Moreover, complete coverage holds: $\bigcup_{k=1}^{K} \mathcal{S}_\omega(v_k) \supseteq \tilde{B}$.

We say there is an *angular separation* if $\cos(b_i, b_k) \le 0$ for any $i, k = 1, \ldots, K$ (i.e., the angles for all pairs are at least $\pi/2$), then $\omega \in \left(1 - \frac{r}{R_{max}}, 1\right) \ne \emptyset$. Thus, under angular separation, the range $\omega$ that allows for full coverage is nonempty independently of $K$. Our result is in agreement with that of Nguyen (2015), whose result suggested that topic simplex $B$ can be consistently estimated without knowing $K$, provided there is a minimum edge length $a_{min} > 0$. The notion of angular separation leads naturally to the Conic Scan-and-Cover algorithm. Before getting there, we show a series of results allowing us to further extend the range of admissible $\omega$.

The inclusion of a spherical ball centered at $C_p$ allows us to expand substantially the range of $\omega$ for which conical coverage continues to hold. In particular, we can reduce the lower bound on $\omega$ in Proposition 1, since we only need to cover the regions near the vertices of $\tilde{B}$ with cones using the following proposition. Fig. 1b provides an illustration.

**Proposition 2.** Let $\mathscr{B}(C_p, \mathcal{R}) = \{\tilde{p} \in \mathbb{R}^V | \|\tilde{p} - C_p\|_2 \le \mathcal{R}\}$, $\mathcal{R} > 0$; $\omega_1, \omega_2$ given in Prop. 1, and

$$\omega_3 := 1 - \min\left\{ \min_{i,k}\left( \frac{R_k \sin^2(b_i, b_k)}{\mathcal{R}} + \cos(b_i, b_k)\sqrt{1 - \frac{R_k^2 \sin^2(b_i, b_j)}{\mathcal{R}^2}} \right), 1 \right\}, \qquad (1)$$

then we have $\bigcup_{k=1}^{K} \mathcal{S}_\omega(v_k) \cup \mathscr{B}(C_p, \mathcal{R}) \supseteq \tilde{B}$ whenever $\omega \in (\min\{\omega_1, \omega_3\}, \omega_2)$.

Notice that as $\mathcal{R} \to R_{max}$, the value of $\omega_3 \to 0$. Hence if $\mathcal{R} \leq R_{min} \approx R_{max}$, the admissible range for $\omega$ in Prop. 2 results in a substantial strengthening from Prop. 1. It is worth noting that the above two geometric propositions do not require any distributional properties inside the simplex.

**Coverage leftovers**  In practice complete coverage may fail if $\omega$ and $\mathcal{R}$ are chosen outside of corresponding ranges suggested by the previous two propositions. In that case, it is useful to note that leftover regions will have a very low mass. Next we quantify the mass inside a cone that *does* contain a vertex, which allows us to *reject* a cone that has low mass, therefore not containing a vertex in it.

**Proposition 3.** The cone $S_\omega(v_1)$ whose axis is a topic direction $v_1$ has mass

$$\mathbb{P}(\mathcal{S}_\omega(v_1)) > \mathbb{P}(\Lambda_c(b_1)) = \frac{\int_{1-c}^1 \theta_1^{\alpha_1-1}(1-\theta_1)^{\sum_{i\neq1}\alpha_i-1}\mathrm{d}\theta_1}{\int_0^1 \theta_1^{\alpha_1-1}(1-\theta_1)^{\sum_{i\neq1}\alpha_i-1}\mathrm{d}\theta_1} =$$

$$\frac{c^{\sum_{i\neq1}\alpha_i}(1-c)^{\alpha_1}\Gamma(\sum_{i=1}^K \alpha_i)}{(\sum_{i\neq1}\alpha_i)\Gamma(\alpha_1)\Gamma(\sum_{i\neq1}\alpha_i)}\left[1 + \frac{c\sum_{i=1}^K \alpha_i}{\sum_{i\neq1}\alpha_i+1} + \frac{c^2(\sum_{i=1}^K \alpha_i)(\sum_{i=1}^K \alpha_i+1)}{(\sum_{i\neq1}\alpha_i+1)(\sum_{i\neq1}\alpha_i+2)} + \cdots\right], \tag{2}$$

where $\Lambda_c(b_1)$ is the simplicial cap of $\mathcal{S}_\omega(v_1)$ which is composed of vertex $b_1$ and a base parallel to the corresponding base of $\tilde{B}$ and cutting adjacent edges of $\tilde{B}$ in the ratio $c : (1-c)$.

See Fig. 1c for an illustration for the simplicial cap described in the proposition. Given the lower bound for the mass around a cone containing a vertex, we have arrived at the following guarantee.

**Proposition 4.** For $\lambda \in (0, 1)$, let $c_\lambda$ be such that $\lambda = \min_k \mathbb{P}(\Lambda_{c_\lambda}(b_k))$ and let $\omega_\lambda$ be such that

$$c_\lambda = \left(\left(2\sqrt{1 - \frac{r^2}{R_{max}^2}}\right)(\sin(d)\cot(\arccos(1-\omega_\lambda)) + \cos(d))\right)^{-1}, \tag{3}$$

where angle $d \leq \min_{i,k} \angle(b_k, b_k - b_i)$. Then, as long as

$$\omega \in \left(\omega_\lambda, \max\left(\frac{a_{min}^2}{2R_{max}^2}, \max_{i,k=1,\ldots,K}(1 - \cos(b_i, b_k))\right)\right), \tag{4}$$

the bound $\mathbb{P}(\mathcal{S}_\omega(v_k)) \geq \lambda$ holds for all $k = 1, \ldots, K$.

### 3.2  CoSAC: Conic Scan-and-Cover algorithm

Having laid out the geometric foundations, we are ready to present the Conic Scan-and-Cover (CoSAC) algorithm, which is a scanning procedure for detecting the presence of simplicial vertices based on data drawn randomly from the simplex. The idea is simple: iteratively pick the farthest point from the center estimate $\hat{C}_p := \frac{1}{M}\sum_m p_m$, say $v$, then construct a cone $\mathcal{S}_\omega(v)$ for some suitably chosen $\omega$, and remove all the data residing in this cone. Repeat until there is no data point left.

Specifically, let $A = \{1, \ldots, M\}$ be the index set of the initially unseen data, then set $v := \operatorname*{argmax}_{\tilde{p}_m:m\in A} \|\tilde{p}_m\|_2$ and update $A := A \setminus \mathcal{S}_\omega(v)$. The parameter $\omega$ needs to be sufficiently large to ensure that the farthest point is a good estimate of a true vertex, and that the scan will be completed in exactly $K$ iterations; $\omega$ needs to be not too large, so that $\mathcal{S}_\omega(v)$ does not contain more than one vertex. The existence of such $\omega$ is guaranteed by Prop. 1. In particular, for an equilateral $\tilde{B}$, the condition of the Prop. 1 is satisfied as long as $\omega \in (1 - 1/\sqrt{K-1}, 1 + 1/(K-1))$.

In our setting, $K$ is unknown. A smaller $\omega$ would be a more robust choice, and accordingly the set $A$ will likely remain non-empty after $K$ iterations. See the illustration of Fig. 1a, where the blue regions correspond to $A$ after $K = 3$ iterations of the scan. As a result, we proceed by adopting a stopping criteria based on Prop. 2: the procedure is stopped as soon as $\forall m \in A \|\tilde{p}_m\|_2 < \mathcal{R}$, which allows us to complete the scan in $K$ iterations (as in Fig. 1b for $K = 3$).

The CoSAC algorithm is formally presented by Algorithm 1. Its running is illustrated in Fig. 2, where we show iterations 1, 26, 29, 30 of the algorithm by plotting norms of the centered documents

in the active set $A$ and cone $\mathcal{S}_\omega(v)$ against cosine distance to the chosen direction of a topic. Iteration 30 (right) satisfies stopping criteria and therefore CoSAC recovered correct $K = 30$. Note that this type of visual representation can be useful in practice to verify choices of $\omega$ and $\mathcal{R}$. The following theorem establishes the consistency of the CoSAC procedure.

**Theorem 1.** Suppose $\{\beta_1, \ldots, \beta_K\}$ are the true topics, incenter $C_p$ is given, $\theta_m \sim \mathrm{Dir}_K(\alpha)$ and $p_m := \sum_k \beta_k \theta_{mk}$ for $m = 1, \ldots, M$ and $\alpha \in \mathbb{R}_+^K$. Let $\hat{K}$ be the estimated number of topics, $\{\hat{\beta}_1, \ldots, \hat{\beta}_{\hat{K}}\}$ be the output of Algorithm 1 trained with $\omega$ and $\mathcal{R}$ as in Prop. 2. Then $\forall\, \epsilon > 0$,

$$
\mathbb{P}\left( \left\{ \min_{j \in \{1, \ldots, \hat{K}\}} \|\beta_i - \hat{\beta}_j\| > \epsilon \,, \text{ for any } i \in \{1, \ldots, \hat{K}\} \right\} \cup \{K \neq \hat{K}\} \right) \to 0 \text{ as } M \to \infty.
$$

**Remark** We found the choices $\omega = 0.6$ and $\mathcal{R}$ to be median of $\{\|\tilde{p}_1\|_2, \ldots, \|\tilde{p}_M\|_2\}$ to be robust in practice and agreeing with our theoretical results. From Prop. 3 it follows that choosing $\mathcal{R}$ as median length is equivalent to choosing $\omega$ resulting in an edge cut ratio $c$ such that $1 - \frac{K}{K-1}(\frac{c}{1-c})^{1-1/K} \geq 1/2$, then $c \leq (\frac{K-1}{2K})^{K/(K-1)}$, which, for any equilateral topic simplex $B$, is satisfied by setting $\omega \in (0.3, 1)$, provided that $K \leq 2000$ based on the Eq. (3).

## 4 Document Conic Scan-and-Cover algorithm

In the topic modeling problem, $p_m$ for $m = 1, \ldots, M$ are *not* given. Instead, under the bag-of-words assumption, we are given the frequencies of words in documents $w_1, \ldots, w_M$ which provide a point estimate $\bar{w}_m := w_m/N_m$ for the $p_m$. Clearly, if number of documents $M \to \infty$ and length of documents $N_m \to \infty\ \forall m$, we can use Algorithm 1 with the plug-in estimates $\bar{w}_m$ in place of $p_m$, since $\bar{w}_m \to p_m$. Moreover, $C_p$ will be estimated by $\hat{C}_p := \frac{1}{M}\sum \bar{w}_m$. In practice, $M$ and $N_m$ are finite, some of which may take relatively small values. Taking the topic direction to be the farthest point in the topic simplex, i.e., $v = \underset{\tilde{w}_m : m \in A}{\mathrm{argmax}} \|\tilde{w}_m\|_2$, where $\tilde{w}_m := \bar{w}_m - \hat{C}_p \in \Delta_0^{V-1}$, may no longer yield a robust estimate, because the variance of this topic direction estimator can be quite high (in the Supplement we show that it is upper bounded with $(1 - 1/V)/N_m$).

To obtain improved estimates, we propose a technique that we call "mean-shifting". Instead of taking the farthest point in the simplex, this technique is designed to shift the estimate of a topic to a high density region, where true topics are likely to be found. Precisely, given a (current) cone $\mathcal{S}_\omega(v)$, we re-position the cone by updating $v := \underset{v}{\mathrm{argmin}} \sum_{m \in \mathcal{S}_\omega(v)} \|\tilde{w}_m\|_2 (1 - \cos(\tilde{w}_m, v))$. In other words, we re-position the cone by centering it around the *mean direction* of the cone weighted by the norms of the data points inside, which is simply given by $v \propto \sum_{m \in \mathcal{S}_\omega(v)} \tilde{w}_m / \mathrm{card}(\mathcal{S}_\omega(v))$. This results in reduced variance of the topic direction estimate, due to the averaging over data residing in the cone.

The mean-shifting technique may be slightly modified and taken as a local update for a subsequent optimization which cycles through the entire set of documents and iteratively updates the cones. The optimization is with respect to the following weighted spherical k-means objective:

$$
\min_{\|v_k\|_2 = 1, k = 1, \ldots K} \sum_{k=1}^{K} \sum_{m \in S^k(v_k)} \|\tilde{w}_m\|_2 (1 - \cos(v_k, \tilde{w}_m)), \tag{5}
$$

where cones $S^k(v_k) = \{m \,|\, d_{\cos}(v_k, \tilde{p}_m) < d_{\cos}(v_l, \tilde{p}_i)\ \forall l \neq k\}$ yield a disjoint data partition $\bigsqcup_{k=1}^{K} S^k(v_k) = \{1, \ldots, M\}$ (this is different from $\mathcal{S}_\omega(v_k)$). The rationale of spherical k-means optimization is to use full data for estimation of topic directions, hence further reducing the variance due to short documents. The connection between objective function (5) and topic simplex estimation is given in the Supplement. Finally, obtain topic norms $R_k$ along the directions $v_k$ using maximum projection: $R_k := \underset{m : m \in S^k(v_k)}{\max} \langle v_k, \tilde{w}_m \rangle$. Our entire procedure is summarized in Algorithm 2.

**Remark** In Step 9 of the algorithm, cone $\mathcal{S}_\omega(v)$ with a very low cardinality, i.e., $\mathrm{card}(\mathcal{S}_\omega(v)) < \lambda M$, for some small constant $\lambda$, is discarded because this is likely an outlier region that does not actually contain a true vertex. The choice of $\lambda$ is governed by results of Prop. 4. For small $\alpha_k = 1/K,\ \forall k,$

$\lambda \leq \mathbb{P}(\Lambda_c) \approx \frac{c^{(K-1)/K}}{(K-1)(1-c)}$ and for an equilateral $\tilde{B}$ we can choose $d$ such that $\cos(d) = \sqrt{\frac{K+1}{2K}}$. Plugging these values into Eq. (3) leads to $c = \left( \left( 2\sqrt{1 - \frac{1}{K^2}} \right) \left( \sqrt{\frac{K-1}{2K}} (\frac{1-\omega}{\sqrt{1-(1-\omega)^2}}) + \sqrt{\frac{K+1}{2K}} \right) \right)^{-1}$.

Now, plugging in $\omega = 0.6$ we obtain $\lambda \leq K^{-1}$ for large $K$. Our approximations were based on large $K$ to get a sense of $\lambda$, we now make a conservative choice $\lambda = 0.001$, so that $(K)^{-1} > \lambda \; \forall K < 1000$. As a result, a topic is rejected if the corresponding cone contains less than 0.1% of the data.

**Finding anchor words using Conic Scan-and-Cover**    Another approach to reduce the noise is to consider the problem from a different viewpoint, where Algorithm 1 will prove itself useful. RecoverKL by Arora et al. (2012) can identify topics with diminishing errors (in number of documents $M$), *provided* that topics contain anchor words. The problem of finding anchor words geometrically reduces to identifying rows of the word-to-word co-occurrence matrix that form a simplex containing other rows of the same matrix (cf. Arora et al. (2012) for details). An advantage of this approach is that noise in the word-to-word co-occurrence matrix goes to zero as $M \to \infty$ no matter the document lengths, hence we can use Algorithm 1 with "documents" being rows of the word-to-word co-occurrence matrix to learn anchor words nonparametrically and then run RecoverKL to obtain topic estimates. We will call this procedure cscRecoverKL.

---

**Algorithm 1** Conic Scan-and-Cover (CoSAC)

---

**Input:** document generating distributions $p_1, \ldots, p_M$,
     angle threshold $\omega$, norm threshold $\mathcal{R}$
**Output:** topics $\beta_1, \ldots, \beta_k$
   1: $\hat{C}_p = \frac{1}{M} \sum_m p_m$ {find center};     $\tilde{p}_m := p_m - \hat{C}_p$ for $m = 1, \ldots, M$ {center the data}
   2: $A_1 = \{1, \ldots, M\}$ {initialize active set};     $k = 1$ {initialize topic count}
   3: **while** $\exists m \in A_k : \|\tilde{p}_m\|_2 > \mathcal{R}$ **do**
   4:     $v_k = \underset{\tilde{p}_m : m \in A_k}{\operatorname{argmax}} \|\tilde{p}_m\|_2$ {find topic}
   5:     $\mathcal{S}_\omega(v_k) = \{m : d_{\cos}(\tilde{p}_m, v_k) < \omega\}$ {find cone of near documents}
   6:     $A_k = A_k \setminus \mathcal{S}_\omega(v_k)$ {update active set}
   7:     $\beta_k = v_k + \hat{C}_p$, $k = k + 1$ {compute topic}
   8: **end while**

---

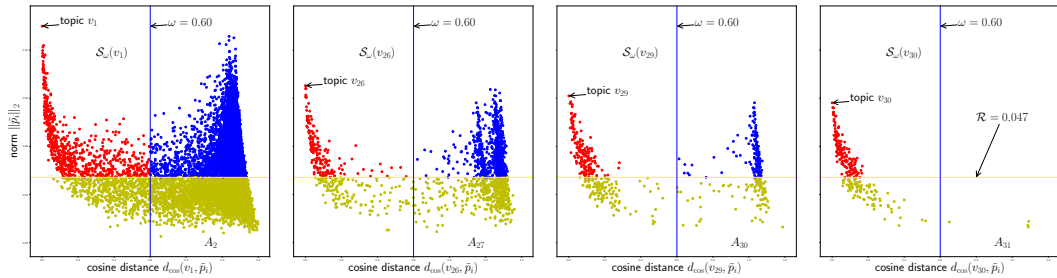

Figure 2: Iterations 1, 26, 29, 30 of the Algorithm 1. Red are the documents in the cone $\mathcal{S}_\omega(v_k)$; blue are the documents in the active set $A_{k+1}$ for next iteration. Yellow are documents $\|\tilde{p}_m\|_2 < \mathcal{R}$.

# 5 Experimental results

## 5.1 Simulation experiments

In the simulation studies we shall compare CoSAC (Algorithm 2) and cscRecoverKL based on Algorithm 1 both of which don't have access to the true $K$, versus popular parametric topic modeling approaches (trained with true $K$): Stochastic Variational Inference (SVI), Collapsed Gibbs sampler, RecoverKL and GDM (more details in the Supplement). The comparisons are done on the basis of minimum-matching Euclidean distance, which quantifies distance between topic simplices (Tang et al., 2014), and running times (perplexity scores comparison is given in the Supplement). Lastly we will demonstrate the ability of CoSAC to recover correct number of topics for a varying $K$.

---

**Algorithm 2** CoSAC for documents

---

**Input:** normalized documents $\bar{w}_1, \ldots, \bar{w}_M$,
    angle threshold $\omega$, norm threshold $\mathcal{R}$, outlier threshold $\lambda$

**Output:** topics $\beta_1, \ldots, \beta_k$

  1: $\hat{C}_p = \frac{1}{M} \sum_m \bar{w}_m$ {find center};      $\tilde{w}_m := \bar{w}_m - \hat{C}_p$ for $m = 1, \ldots, M$ {center the data}

  2: $A_1 = \{1, \ldots, M\}$ {initialize active set};     $k = 1$ {initialize topic count}

  3: **while** $\exists\, m \in A_k : \|\tilde{w}_m\|_2 > \mathcal{R}$ **do**

  4:    $v_k = \underset{\tilde{w}_m : m \in A_k}{\text{argmax}} \|\tilde{w}_m\|_2$ {initialize direction}

  5:    **while** $v_k$ not converged **do** {mean-shifting}

  6:      $\mathcal{S}_\omega(v_k) = \{m : d_{\cos}(\tilde{w}_m, v_k) < \omega\}$ {find cone of near documents}

  7:      $v_k = \sum_{m \in \mathcal{S}_\omega(v_k)} \tilde{w}_m / \,\text{card}(\mathcal{S}_\omega(v_k))$ {update direction}

  8:    **end while**

  9:    $A_k = A_k \setminus \mathcal{S}_\omega(v_k)$ {update active set}
        **if** $\text{card}(\mathcal{S}_\omega(v_k)) > \lambda M$    **then** $k = k + 1$ {record topic direction}

10: **end while**

11: $v_1, \ldots, v_k =$ weighted spherical k-means $(v_1, \ldots, v_k, \tilde{w}_1, \ldots, \tilde{w}_M)$

12: **for** $l$ in $\{1, \ldots, k\}$ **do**

13:    $R_l := \underset{m : m \in S^l(v_l)}{\max} \langle v_l, \tilde{w}_m \rangle$ {find topic length along direction $v_l$}

14:    $\beta_l = R_l v_l + \hat{C}_p$ {compute topic}

15: **end for**

---

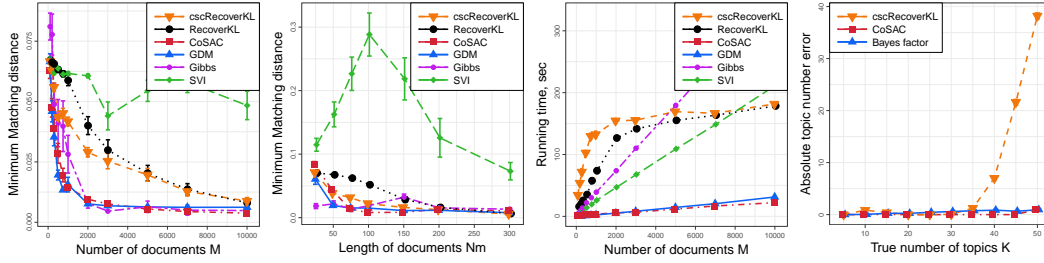

Figure 3: Minimum matching Euclidean distance for (a) varying corpora size, (b) varying length of documents; (c) Running times for varying corpora size; (d) Estimation of number of topics.

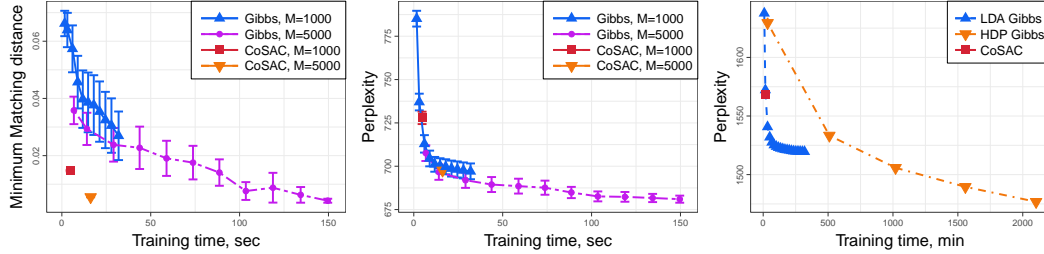

Figure 4: Gibbs sampler convergence analysis for (a) Minimum matching Euclidean distance for corpora sizes 1000 and 5000; (b) Perplexity for corpora sizes 1000 and 5000; (c) Perplexity for NYTimes data.

**Estimation of the LDA topics** First we evaluate the ability of CoSAC and cscRecoverKL to estimate topics $\beta_1, \ldots, \beta_K$, fixing $K = 15$. Fig. 3(a) shows performance for the case of fewer $M \in [100, 10000]$ but longer $N_m = 500$ documents (e.g. scientific articles, novels, legal documents). CoSAC demonstrates performance comparable in accuracy to Gibbs sampler and GDM.

Next we consider larger corpora $M = 30000$ of shorter $N_m \in [25, 300]$ documents (e.g. news articles, social media posts). Fig. 3(b) shows that this scenario is harder and CoSAC matches the performance of Gibbs sampler for $N_m \geq 75$. Indeed across both experiments CoSAC only made mistakes in terms of $K$ for the case of $N_m = 25$, when it was underestimating on average by 4 topics

and for $N_m = 50$ when it was off by around 1, which explains the earlier observation. Experiments with varying $V$ and $\alpha$ are given in the Supplement.

It is worth noting that cscRecoverKL appears to be strictly better than its predecessor. This suggests that our procedure for selection of anchor words is more accurate in addition to being nonparametric.

**Running time**    A notable advantage of the CoSAC algorithm is its speed. In Fig. 3(c) we see that Gibbs, SVI, GDM and CoSAC all have linear complexity growth in $M$, but the slopes are very different and approximately are $IN_m$ for SVI and Gibbs (where $I$ is the number of iterations which has to be large enough for convergence), number of k-means iterations to converge for GDM and is of order $K$ for the CoSAC procedure making it the fastest algorithm of all under consideration.

Next we compare CoSAC to per iteration quality of the Gibbs sampler trained with 500 iterations for $M = 1000$ and $M = 5000$. Fig. 4(b) shows that Gibbs sampler, when true $K$ is given, can achieve good perplexity score as fast as CoSAC and outperforms it as training continues, although Fig. 4(a) suggests that much longer training time is needed for Gibbs sampler to achieve good topic estimates and small estimation variance.

**Estimating number of topics**    Model selection in the LDA context is a quite challenging task and, to the best of our knowledge, there is no "go to" procedure. One of the possible approaches is based on refitting LDA with multiple choices of $K$ and using Bayes Factor for model selection (Griffiths & Steyvers, 2004). Another option is to adopt the Hierarchical Dirichlet Process (HDP) model, but we should understand that it is not a procedure to estimate $K$ of the LDA model, but rather a particular prior on the number of topics, that assumes $K$ to grow with the data. A more recent suggestion is to slightly modify LDA and use Bayes moment matching (Hsu & Poupart, 2016), but, as can be seen from Figure 2 of their paper, estimation variance is high and the method is not very accurate (we tried it with true $K = 15$ and it took above 1 hour to fit and found 35 topics). Next we compare Bayes factor model selection versus CoSAC and cscRecoverKL for $K \in [5, 50]$. Fig. 3(d) shows that CoSAC consistently recovers *exact* number of topics in a wide range.

We also observe that cscRecoverKL does not estimate $K$ well (underestimates) in the higher range. This is expected because cscRecoverKL finds the number of anchor words, *not* topics. The former is decreasing when later is increasing. Attempting to fit RecoverKL with more topics than there are anchor words might lead to deteriorating performance and our modification can address this limitation of the RecoverKL method.

## 5.2   Real data analysis

In this section we demonstrate CoSAC algorithm for topic modeling on one of the standard bag of words datasets — NYTimes news articles. After preprocessing we obtained $M \approx 130,000$ documents over $V = 5320$ words. Bayes factor for the LDA selected the smallest model among $K \in [80, 195]$, while CoSAC selected 159 topics. We think that disagreement between the two procedures is attributed to the misspecification of the LDA model when real data is in play, which affects Bayes factor, while CoSAC is largely based on the geometry of the topic simplex.

The results are summarized in Table 1 — CoSAC found 159 topics in less than 20min; cscRecoverKL estimated the number of anchor words in the data to be 27 leading to fewer topics. Fig. 4(c) compares CoSAC perplexity score to per iteration test perplexity of the LDA (1000 iterations) and HDP (100 iterations) Gibbs samplers. Text files with top 20 words of all topics are included in the Supplementary material. We note that CoSAC procedure recovered meaningful topics, contextually similar to LDA and HDP (e.g. elections, terrorist attacks, Enron scandal, etc.) and also recovered more specific topics about Mike Tyson, boxing and case of Timothy McVeigh which were present among HDP topics, but not LDA ones. We conclude that CoSAC is a practical procedure for topic modeling on large scale corpora able to find meaningful topics in a short amount of time.

## 6   Discussion

We have analyzed the problem of estimating topic simplex without assuming number of vertices (i.e., topics) to be known. We showed that it is possible to cover topic simplex using two types of geometric shapes, cones and a sphere, leading to a class of Conic Scan-and-Cover algorithms. We

Table 1: Modeling topics of NYTimes articles

|  | $K$ | Perplexity | Coherence | Time |
|---|---|---|---|---|
| cscRecoverKL | 27 | 2603 | -238 | 37 min |
| HDP Gibbs | $221 \pm 5$ | $1477 \pm 1.6$ | $-442 \pm 1.7$ | 35 hours |
| LDA Gibbs | 80 | $1520 \pm 1.5$ | $-300 \pm 0.7$ | 5.3 hours |
| CoSAC | 159 | 1568 | -322 | 19 min |

then proposed several geometric correction techniques to account for the noisy data. Our procedure is accurate in recovering the true number of topics, while remaining practical due to its computational speed. We think that angular geometric approach might allow for fast and elegant solutions to other clustering problems, although as of now it does not immediately offer a unifying problem solving framework like MCMC or variational inference. An interesting direction in a geometric framework is related to building models based on geometric quantities such as distances and angles.

## Acknowledgments

This research is supported in part by grants NSF CAREER DMS-1351362, NSF CNS-1409303, a research gift from Adobe Research and a Margaret and Herman Sokol Faculty Award.

## Footnotes

[1]Code is available at https://github.com/moonfolk/Geometric-Topic-Modeling.

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
