[Supplementary Material]

# Supplementary material for Conic Scan-and-Cover algorithms for nonparametric topic modeling

**Mikhail Yurochkin**
Department of Statistics
University of Michigan
moonfolk@umich.edu

**Aritra Guha**
Department of Statistics
University of Michigan
aritra@umich.edu

**XuanLong Nguyen**
Department of Statistics
University of Michigan
xuanlong@umich.edu

## 1 Proofs of main theorems

We start by reminding the reader of our geometric setup. First, topic simplex $B := \text{Conv}(\beta_1, \ldots, \beta_K)$ is centered at a point denoted by $C_p$. Let $\Delta_0^{V-1} := \{x \in \mathbb{R}^V : x + C_p \in \Delta^{V-1}\}$ — centered probability simplex. Then, write $b_k := \beta_k - C_p \in \Delta_0^{V-1}$ for $k = 1, \ldots, K$ and $\tilde{p}_m := p_m - C_p \in \Delta_0^{V-1}$ for $m = 1, \ldots, M$. Note that re-centering leaves corresponding barycentric coordinates $\theta_m \in \Delta^{K-1}$ unchanged. Moreover, the extreme points of centered topic simplex $\tilde{B} := \text{Conv}\{b_1, \ldots, b_K\}$ can now be represented by their directions $v_k \in \mathbb{R}^V$ and corresponding radii $R_k \in \mathbb{R}_+$ such that $b_k = R_k v_k$ for any $k = 1, \ldots, K$.

### 1.1 Coverage of the topic simplex

Suppose that $C_p$ is the incenter of the topic simplex $\tilde{B}$, with $r$ being the inradius. Recall that the incenter and inradius correspond to the maximum volume sphere inside $\tilde{B}$. Let $a_{i,k}$ denote the distance between the $i^{th}$ and $k^{th}$ vertex of $\tilde{B}$, with $a_{min} \leq a_{i,k} \leq a_{max}$ for all $i, k$, and $R_{max}, R_{min}$ such that $R_{min} \leq R_k := \|b_k\|_2 \leq R_{max} \; \forall \; k = 1, \ldots, K$

**Proposition 1.** For simplex $\tilde{B}$ and $\omega \in (\omega_1, \omega_2)$, where $\omega_1 = 1 - r/R_{max}$ and $\omega_2 = \max\{(a_{min}^2)/(2R_{max}^2), \max_{i,k=1,\ldots,K}(1 - \cos(b_i, b_k)\}$, the cone $\mathcal{S}_\omega(v)$ around any vertex direction $v$ of $\tilde{B}$ contains exactly one vertex. Moreover, complete coverage holds: $\bigcup_{k=1}^{K} \mathcal{S}_\omega(v_k) \supseteq \tilde{B}$.

*Proof.* Let $\omega_0 = \frac{a_{min}^2}{2R_{max}^2}$. Then, for any $k \in \{1, \ldots, K\}$, for any $\omega \leq \omega_0$, $\mathcal{S}_\omega(v_k)$ does not contain any other vertices. This can be explained as follows. Fix $k$, and choose $i \in \{1, \ldots, K\} \neq k$. Define $\phi_{i,k}$ as the angle at $C_p$ made by the side connecting the vertex $i$ and vertex $k$. Then from the cosine law for triangles, we have

$$\cos(\phi_{i,k}) = \frac{R_i^2 + R_k^2 - a_{i,k}^2}{2R_i R_k}.$$

Now, for any $\phi \leq \min_{i,k} \phi_{i,k}$, with $\omega_\phi = 1 - \cos(\phi)$, the cone $\mathcal{S}_{\omega_\phi}(v_k)$ does not cover any vertex other than vertex $k$, for any $k$. Now $\phi_1 = \min_{i,k} \phi_{i,k}$ satisfies

$$1 - \cos(\phi_1) \leq \frac{a_{min}^2}{2R_{max}^2} - \frac{(R_{max} - R_{min})^2}{2R_{max}R_{min}} \leq \frac{a_{min}^2}{2R_{max}^2}.$$

from which we obtain the upper bound for $\omega$. For the lower bound, consider for vertex $k$, $\mathcal{S}(v_k)$ the cone connecting the incenter to facial incenters of facets containing vertex $k$. Then $\bigcup_{k=1}^{K} \mathcal{S}(v_k) \supseteq \tilde{B}$.

Figure 1: $C : k^{th}$ vertex point, $A$ : point where the adjacent side to the vertex has been cut off by the sphere, $R_k$: distance to $k^{th}$ vertex from incenter, $\mathcal{R}$ : radius of sphere, $B$ : incenter

Now for each $k$, $\mathcal{S}(v_k) \subseteq \mathcal{S}_{\omega_2}(v_k)$, where $\omega_2 = 1 - \cos(\phi_2)$, with $\phi_2$ satisfying $\cos(\phi_2) \leq \min_{k \in \{1,\dots,K\}} \frac{r}{R_k}$. From this we get the lower bound. The restriction $2R_{max}^2 \leq a_{min}^2$ is needed to ensure that the set $\left\{\omega : 1 - (\frac{r}{R_{max}}) \leq \omega \leq (\frac{a_{min}^2}{2R_{max}^2})\right\}$ is non-empty. $\qquad\square$

**Proposition 2.** Let $\mathscr{B}(C_p, \mathcal{R}) = \{\tilde{p} \in \mathbb{R}^V \mid \|\tilde{p} - C_p\|_2 \leq \mathcal{R}\}$, $\mathcal{R} > 0$; $\omega_1, \omega_2$ given in Prop. 1, and

$$\omega_3 := 1 - \min\left\{\min_{i,k}\left(\frac{R_k \sin^2(b_i, b_k)}{\mathcal{R}} + \cos(b_i, b_k)\sqrt{1 - \frac{R_k^2 \sin^2(b_i, b_j)}{\mathcal{R}^2}}\right), 1\right\}, \qquad (1)$$

then we have $\bigcup_{k=1}^{K} \mathcal{S}_{\omega}(v_k) \cup \mathscr{B}(C_p, \mathcal{R}) \supseteq \tilde{B}$ whenever $\omega \in (\min\{\omega_1, \omega_3\}, \omega_2)$.

*Proof.* Let $\phi_{i,k} = \arccos(1 - \omega_{i,k})$ be the angle formed by the line joining the $k^{th}$ vertex to the incenter $C_p$ and the radial vector from incenter to the point where the sphere cuts the edge connecting $i$ and $k$ (segment $AB$ on Fig. 1). From the sine law for a triangle we have

$$\cos(\phi_{i,k}) + \cot(b_i, b_k)\sin(\phi_{i,k}) - \frac{R_k}{\mathcal{R}} = 0. \qquad (2)$$

Solving for $\phi_{i,k}$ we have $\cos(\phi_{i,k}) = \left(\frac{R_k \sin^2(b_i, b_k)}{\mathcal{R}} + \cos(b_i, b_k)\sqrt{1 - \frac{R_k^2 \sin^2(b_i, b_k)}{\mathcal{R}^2}}\right)$. Now, since we must choose the largest such $\phi$ over all $i$ and $k$, the bound follows immediately. Notice that as $\mathcal{R} \to R_{max}$, the value of $\left(\frac{R_k \sin^2(b_i, b_k)}{\mathcal{R}} + \cos(b_i, b_k)\sqrt{1 - \frac{R_k^2 \sin^2(b_i, b_k)}{\mathcal{R}^2}}\right) \to 1$, whereas $\frac{r}{R_{max}} < 1$ strictly. Thus, as $\mathcal{R}$ increases the lower bound in this limiting scenario is dominated by $1 - \min_{i,k}\left(\frac{R_k \sin^2(b_i, b_k)}{\mathcal{R}} + \cos(b_i, b_k)\sqrt{1 - \frac{R_k^2 \sin^2(b_i, b_k)}{\mathcal{R}^2}}\right)$, thereby obtaining an improvement in the bound from Proposition 1. $\qquad\square$

**Proposition 3.** The cone $\mathcal{S}_{\omega}(v_1)$ whose axis is a topic direction $v_1$ has mass

$$\mathbb{P}(\mathcal{S}_{\omega}(v_1)) > \mathbb{P}(\Lambda_c(b_1)) = \frac{\int_{1-c}^{1} \theta_1^{\alpha_1-1}(1-\theta_1)^{\sum_{i\neq 1}\alpha_i-1}\mathrm{d}\theta_1}{\int_0^1 \theta_1^{\alpha_1-1}(1-\theta_1)^{\sum_{i\neq 1}\alpha_i-1}\mathrm{d}\theta_1} =$$

$$\frac{c^{\sum_{i\neq 1}\alpha_i}(1-c)^{\alpha_1}\Gamma(\sum_{i=1}^{K}\alpha_i)}{(\sum_{i\neq 1}\alpha_i)\Gamma(\alpha_1)\Gamma(\sum_{i\neq 1}\alpha_i)}\left[1 + \frac{c\sum_{i=1}^{K}\alpha_i}{\sum_{i\neq 1}\alpha_i+1} + \frac{c^2(\sum_{i=1}^{K}\alpha_i)(\sum_{i=1}^{K}\alpha_i+1)}{(\sum_{i\neq 1}\alpha_i+1)(\sum_{i\neq 1}\alpha_i+2)} + \cdots\right], \qquad (3)$$

where $\Lambda_c(b_1)$ is the simplicial cap of $\mathcal{S}_\omega(v_1)$ which is composed of vertex $b_1$ and a base parallel to the corresponding base of $\tilde{B}$ and cutting adjacent edges of $\tilde{B}$ in the ratio $c : (1-c)$.

The truncated beta probability calculations in Proposition 3 can be found in Olver et al. (2010).

**Proposition 4.** For $\lambda \in (0,1)$, let $c_\lambda$ be such that $\lambda = \min_k \mathbb{P}(\Lambda_{c_\lambda}(b_k))$ and let $\omega_\lambda$ be such that

$$c_\lambda = \left( \left( 2\sqrt{1 - \frac{r^2}{R_{max}^2}} \right) (\sin(d)\cot(\arccos(1-\omega_\lambda)) + \cos(d)) \right)^{-1}, \qquad (4)$$

where angle $d \leq \min_{i,k} \angle(b_k, b_k - b_i)$. Then, as long as

$$\omega \in \left( \omega_\lambda, \max\left( \frac{a_{min}^2}{2R_{max}^2}, \max_{i,k=1,\dots,K}(1 - \cos(b_i, b_k)) \right) \right), \qquad (5)$$

the bound $\mathbb{P}(\mathcal{S}_\omega(v_k)) \geq \lambda$ holds for all $k = 1, \dots, K$.

*Proof.* Consider Figure 1, with length of $AC = a_{i,k}c$, where $c$ is the proportion in which the cone cuts $AC$, the edge joining vertex $i$ and vertex $k$. Now, from the sine law of a triangle,

$$\frac{R_k}{a_{i,k}c} = \sin(b_i, b_k)\cot\phi_{i,k} + \cos(b_i, b_k) \qquad (6)$$

where $\phi_{i,k}$ is as defined in the proof of Proposition 2. Now $\frac{a_{i,k}}{R_k} \leq \frac{2(\sqrt{R_{max}^2 - r^2})}{R_{max}}$. The choice of $\phi_\lambda = \cos\omega_\lambda$ satisfies

$$c_\lambda \geq \frac{1}{2\sqrt{1 - \frac{r^2}{R_{max}^2}}} \min_{i,k} \frac{1}{\sin(b_i, b_k)\cot\phi_\lambda + \cos(b_i, b_k)} \qquad (7)$$

therefore proves the theorem. Since, $\phi_\lambda \leq \frac{\pi}{2} - \angle(b_i, b_k)$, for all $i, k$, the function $\sin(b_i, b_k)\cot\phi_\lambda + \cos(b_i, b_k)$ is increasing as the angle between $b_i$ and $b_k$ increases, as can be checked for maxima by the first derivative rule. Using the cosine law,

$$\cos(b_i, b_k) = \frac{-R_i^2 + R_k^2 + a_{i,k}^2}{2a_{i,k}R_k}. \qquad (8)$$

Minimizing this quantity with respect to $i$ and $k$ we get the result. $\qquad\square$

## 1.2 Consistency of the Conic Scan-and-Cover algorithm

Under the LDA setup (as presented in Section 2 of the main text), recall that $a_{i,k}$ is the length of the edge connecting the $i^{th}$ and $k^{th}$ vertex, i.e., $\|\beta_i - \beta_k\|_2 = a_{i,k}$, where $\|\cdot\|_2$ is the $\ell_2$ norm. Let $\mathcal{B}(\cdot, \epsilon)$ denote an $\epsilon$-ball in $\ell_2$-norm. Then the following result states that with high probability there exists a document in a neighborhood of every vertex.

**Lemma 1.** Let $p_m := \sum_k \beta_k \theta_{mk}$ for $m = 1, \dots, M$ as before. Then for any $i$ and any $0 < \epsilon < \max_{k \neq i} a_{i,k}$,

$$\mathbb{P}(p_m \notin \mathcal{B}(\beta_i, \epsilon) \; \forall \, m \in \{1, \dots, M\}) \leq \left( \frac{\int_0^{1 - (\epsilon/\max_{k \neq i} a_{i,k})} \theta_i^{\alpha_i - 1}(1 - \theta_i)^{\sum_{j \neq i} \alpha_j - 1} d\theta_i}{\int_0^1 \theta_i^{\alpha_i - 1}(1 - \theta_i)^{\sum_{j \neq 1} \alpha_j - 1} d\theta_i} \right)^M. \qquad (9)$$

Since $\left( \frac{\int_0^{1 - (\epsilon/\max_{k \neq i} a_{i,k})} \theta_i^{\alpha_i - 1}(1 - \theta_i)^{\sum_{j \neq i} \alpha_j - 1} d\theta_i}{\int_0^1 \theta_i^{\alpha_i - 1}(1 - \theta_i)^{\sum_{j \neq 1} \alpha_j - 1} d\theta_i} \right) < 1$, for all $i$ because Beta distribution is absolutely continuous in $(0, 1)$, the bound on the right hand side goes to 0 as $M \to \infty$.

Let $\{\hat{\beta}_1, \dots, \hat{\beta}_K\}$ be the topics identified by Conic Scan-and-Cover algorithm, with labels permuted according to the minimum matching distance criteria, with $\{\beta_1, \dots, \beta_K\}$ being the true topics. Then the following result shows the consistency of the identified topics.

**Theorem 1.** Suppose $\{\beta_1, \ldots, \beta_K\}$ are the true topics, incenter $C_p$ is given, $\theta_m \sim \mathrm{Dir}_K(\alpha)$ and $p_m := \sum_k \beta_k \theta_{mk}$ for $m = 1, \ldots, M$ and $\alpha \in \mathbb{R}_+^K$. Let $\{\hat{\beta}_1, \ldots, \hat{\beta}_{\hat{K}}\}$ be the output of the Conic Scan-and-Cover algorithm trained with $\omega$ and $\mathcal{R}$ as in Proposition 2. Then $\forall \epsilon > 0$,

$$\mathbb{P}\left(\left\{\min_{j \in \{1, \ldots, \hat{K}\}} \|\beta_i - \hat{\beta}_j\| > \epsilon, \text{ for any } i \in \{1, \ldots, \hat{K}\}\right\} \cup \{K \neq \hat{K}\}\right) \to 0 \text{ as } M \to \infty.$$

*Proof.* From the description of the Conic Scan-and-Cover algorithm it suffices to prove that for the suitable choice of $\omega, \mathcal{R}$ as in Proposition 2 there holds $\mathbb{P}(\exists \, x_i \in \{p_1, \ldots, p_m\}$ such that $\|\beta_i - x_i\| < \epsilon \, \forall \, i \in \{1, \ldots, K\}) \to 1$ as $M \to \infty$. But this probability expression is bounded from below by $1 - \sum_{i=1}^K \mathbb{P}(p_m \notin \mathscr{B}(\beta_i, \epsilon) \, \forall \, m \in \{1, \ldots, M\})$. The conclusion now follows from Lemma 1. $\qquad\square$

### 1.3 Variance argument for multinomial setup

In the topic modeling problem we are not given $p_m$ for $m = 1, \ldots, M$. Under the bag-of-words assumption we have access to the frequencies of words in documents $w_1, \ldots, w_M$ which provide a point estimate $\bar{w}_m := w_m/N_m$ for the $p_m$. The following proposition establishes a bound on the variation of $\bar{w}_m$ from $p_m$.

**Proposition 5.**

$$\mathbb{E}[\|\bar{w}_m - p_m\|_2^2] \leq \frac{1 - (1/V)}{N_m}. \tag{10}$$

*Proof.* By iterated expectation identity,

$$\mathbb{E}[\|\bar{w}_m - p_m\|_2^2] = \mathbb{E}\left[\mathbb{E}\left[\sum_{i=1}^V \|\bar{w}_{mi} - p_{mi}\|_2^2 \Big| p_m\right]\right]$$

$$= \mathbb{E}\left[\sum_{i=1}^V \frac{p_{mi}(1 - p_{mi})}{N_m}\right]$$

$$= \frac{1 - \mathbb{E}[\sum_{i=1}^V p_{mi}^2]}{N_m} \leq \frac{1 - (1/V)}{N_m}.$$

The second equality follows because conditioned on $p_m$, each $w_{mi} \sim \mathrm{Bin}(N_m, p_{mi})$. The last inequality follows from Cauchy-Schwartz Inequality. $\qquad\square$

## 2 Spherical k-means for topic modeling

We aim to clarify the role of Step 11 of the document Conic Scan-and-Cover algorithm, a geometric correction technique based on weighted spherical k-means optimization.

### 2.1 Topic directions as solutions to weighted spherical k-means

Let centered document norms $r_m := \|\tilde{p}_m\|_2$ for $m = 1, \ldots, M$ and $\alpha_k(v) := \cos(b_k, v)$, cosine of the angle between direction $v$ and $k$-th topic. The weighted spherical k-means objective takes the form

$$\min_{\|v_k\|_2 = 1, k = 1, \ldots K} \sum_{k=1}^K \sum_{m \in S^k(v_k)} r_m(1 - \cos(v_k, \tilde{p}_m)), \tag{11}$$

where $S^k(v_k) := \{m | \cos(v_k, \tilde{p}_m)) > \cos(v_l, \tilde{p}_m) \, \forall l \neq k\}$. Next observe that:

$$r_m \cos(v_k, \tilde{p}_m) = \langle v_k, \tilde{p}_m \rangle = \sum_{i=1}^K \theta_{mi} \langle v_k, b_i \rangle = \sum_{i=1}^K \theta_{mi} R_i \alpha_i(v_k), \tag{12}$$

so our objective 11 becomes:

$$\max_{\|v_k\|_2 = 1, k = 1, \ldots K} \sum_{k=1}^K \sum_{m \in S^k(v_k)} \sum_{i=1}^K \theta_{mi} R_i \alpha_i(v_k). \tag{13}$$

Now, if $R_1 = \ldots = R_K$ and $\alpha_i(b_k) = \alpha_i(b_l) \ \forall \ k, l \neq i$, which implies that topic simplex is equilateral, we see that cluster boundaries of topic directions are given by $m \in S^k(b_k)$ iff $\theta_{mk} > \theta_{ml} \ \forall \ l \neq k$. Observe that the corresponding partition is defined by the geometric *medians* of topic simplex, which in turn partitions it into equal volume parts. Then, assuming that the topic simplex $B$ is symmetric, combined with the symmetricity of the Dirichlet distribution of $\theta_m$-s, it follows that $b_k$ is the centroid of $S^k(b_k)$ for $k = 1, \ldots, K$.

## 2.2 Role of the spherical k-means in CoSAC algorithm for documents

The result of Section 2.1 shows that weighted spherical k-means with Lloyd type updates (Lloyd, 1982) will converge to the directions of the true topics if it is initialized in their respective neighborhoods and equilaterality of $B$ and symmetricity of Dirichlet for document topic proportions is satisfied.

Recall that goal of the Conic Scan-and-Cover is to find the number of topics and their directions, while Mean Shifting was used to address the noise in the data. We proceed to compare weighted spherical k-means by itself (with 500 iterations, which makes it slower than CoSAC) versus document Conic Scan-and-Cover with only Mean Shifting and the full document Conic Scan-and-Cover algorithm to see the effect of the spherical k-means post-processing step. Results in Fig. 2 are for the same scenarios as in the main text – that is when either documents are short $N_m \in [25, 300]$ but corpora is large $M = 30000$ or when documents are longer $N_m = 500$ and corpora is smaller $M \in [100, 10000]$. We see that spherical k-means by itself does not succeed, whereas when used as a postprocessing step for CoSAC it allows for a slight improvement when documents are short. This is because it operates on the full data partition when taking averages for direction estimation, while Mean Shifting only has access to the data in its respective cone $S_\omega(v)$. Using more data is important for noise reduction when documents are short as suggested by our analysis.

Figure 2: Minimum matching Euclidean distance for (a) varying corpora size, (b) varying length of documents. Perplexity for (c) varying corpora sizes, (d) varying length of documents.

# 3 Additional experiments

## 3.1 Perplexity comparison

In this section we present perplexity scores comparison for the same experiments as in the main text. For simulation experiments we used $V = 2000$, symmetric $\alpha = \eta = 0.1$. To compute held-out perplexity for the CoSAC we employed projection based estimates for topic proportions $\theta_m$ from Yurochkin & Nguyen (2016), which led to a slightly worse perplexity scores for CoSAC and GDM in comparison to Gibbs sampler. However, CoSAC (except for $N_m = 25$, when it slightly underestimates $K$) shows competitive performance without requiring $K$ as an input. We note that as before cscRecoverKL outperforms RecoverKL in all cases.

## 3.2 Varying vocabulary size $V$

Our next experiment investigates the influence of vocabulary size $V$. We set $N_m = 500$, $M = 5000$, $K = 15$, symmetric $\alpha = \eta = 0.1$ and varied $V$ from 2000 to 15000. We discovered that $\omega = 0.6$ is too small for $V > 10000$, meaning that CoSAC algorithm does not find enough documents in the corresponding cones and keeps discarding without recording topics (per Step 9 of Algorithm 2). This can be explained by the fact that vectors tending to be far apart in high dimensions and relatively (to $V > 10000$) small values of corpora size $M$ and document lengths $N_m$. On the other hand, setting

$\omega = 0.75$ worked well for all values of $V$ in this experiment. Results are reported in Fig. 3(c), (d) and Fig. 4(d). Document CoSAC with $\omega = 0.75$ recovered true $K = 15$ for all values of $V$ and showed better recovery than GDM and Gibbs sampler in terms of minimum matching distance, while Gibbs sampler had slightly better perplexity for higher values of $V$. It is worth reminding that unlike CoSAC, both GDM and the Gibbs sampling based method requires the number of topics $K$ be given.

### 3.3 Impact of $\alpha$

Recall that, per the LDA model, topic proportions $\theta \sim \mathrm{Dir}_K(\alpha)$. Cases with $\alpha > 1$ were previously shown (Nguyen, 2015) to exhibit slower convergence rates of the LDA's posterior estimation (via Gibbs sampler, for instance). Geometrically, large $\alpha$ implies that documents are more likely concentrated near the center of the topic simplex, leaving fewer documents near the vertices; this entails that geometric inference is more challenging. In our choices for parameters $\omega, \mathcal{R}, \lambda$ we relied on small values of $\alpha$ as a more practical scenario. Specifically, we considered $\omega = 0.8$ for this experiment to achieve full coverage of the topic simplex. In our previous experiments we set $\alpha = 0.1$. Now, we consider a larger range, $\alpha \in [0.01, 1.5]$, to gauge its impact more fully. Results are reported in Fig. 4(a), (b) and (c). For smaller values of $\alpha$ CoSAC is demonstrated to be the best algorithm of all under consideration. As $\alpha$ increases, CoSAC can still recover correct $K$ with high accuracy, although the quality of topic estimates deteriorates faster than for Gibbs sampler and GDM. We think that further work on estimation procedures for topic radii $R_k$s (recall that topics are estimated as direction and length along this direction $b_k = R_k v_k$) might address this issue. In this work we considered maximum projection (Step 13 of Algorithm 2 of the main text) to estimate $R_k$s, which might not be as accurate when documents are mostly near the center of the topic simplex (i.e., for higher $\alpha$).

Figure 3: Perplexity for (a) varying corpora size, (b) varying length of documents, (c) varying vocabulary size; (d) Minimum matching Euclidean distance for varying vocabulary size.

Figure 4: Varying $\alpha$ (a) Minimum matching Euclidean distance, (b) Perplexity, (c) Estimation of number of topics; (d) Estimation of number of topics for varying vocabulary size.

## 4 Implementation details

In this section we give details about the implementations of the algorithms used in simulation studies and real data. We implemented Conic Scan-and-Cover (CoSAC) algorithm in Python with the help of Scipy (Jones et al., 2001–) sparse matrix modules. Geometric Dirichlet Means (GDM) (Yurochkin & Nguyen, 2016) was implemented with the help of Scikit-learn (Pedregosa et al., 2011) k-means implementation (with 10 restarts to avoid local minima of k-means) combined with a geometric correction technique. Codes for CoSAC and GDM are available at `https://github.com/moonfolk/Geometric-Topic-Modeling`. For RecoverKL (Arora et al.,

2012) we applied code from one of the coauthors. To implement cscRecoverKL we used our CoSAC implementation (Algorithm 1 of the main text with outlier threshold $\lambda$ as in Algorithm 2) to find anchor words and then recovery routine from the aforementioned code. For the Gibbs sampling (Griffiths & Steyvers, 2004) we used an lda package in Python that utilizes Cython to achieve C speed. Gibbs sampler was trained with $\alpha = 0.1$, $\eta = 0.01$ and 500 iterations in simulations studies and $\alpha = 0.1$, $\eta = 0.1$ and 1000 iterations in the NYTimes articles[1] analysis. For the SVI (Hoffman et al., 2013) we used Gensim implementation (Řehůřek & Sojka, 2010) with automatic hyperparameters estimation, 50 iterations and 10 passes. Finally for HDP (Teh et al., 2006) we used C++ implementation with default hyperparameter settings and 100 iterations. For all experiments (except large vocabulary sizes and bigger $\alpha$), per discussions in Sections 3.2 and 4 of the main text, parameters of the CoSAC were set to $\omega = 0.6, n = 0.001M$ and $\mathcal{R}$ as median of the centered and normalized document norms. Spherical k-means post-processing step was run for 30 iterations. For cscRecoverKL we set $\omega = 0.4, \lambda = 0.015$ ($\lambda = 0.005$ for real data) and $\mathcal{R}$ as corresponding median of the norms. Note that cscRecoverKL takes word-to-word co-occurrence matrix as input, therefore sample size is $V$ and "documents" are rows of this matrix. Exploring distributional properties of the simplex spanned by the anchor words is outside of the scope of this work, therefore parameter choices were made empirically based on the visual analysis illustrated by Fig. 2 of the main text. All simulated results are reported after 20 repetitions of the data generation for each scenario and NYTimes results for LDA and HDP are reported over 10 refits of the corresponding Gibbs samplers.

## Footnotes

[1] https://archive.ics.uci.edu/ml/datasets/bag+of+words