[Reviews · NeurIPS 2017]

Reviewer 1



This paper presents a geometric algorithm for parameter estimation in LDA when the number of topics is unknown. The method obtains impressive results—its predictive performance on synthetic data is comparable to a Gibbs sampler where the true number of topics is fixed. Moreover the algorithm appears to scale very well with the number of documents (like all spectral methods). Unfortunately, the meat of the paper is dense and difficult to parse. The paper attempts to provide some geometric intuition — but the lack of a clean and well-described picture makes the scattered references to the geometric story (i.e., "conic scans") hard to follow. The caption for figure 1 does not say enough to clearly explain what is being depicted. I don’t have much familiarity with the relevant background literature on geometric algorithms for LDA inference and will defer to other reviewers with stronger background. But I think the general NIPS reader will have too much difficulty reading this paper.

Reviewer 2



The paper proposes a nonparametric topic model based on geometric properties of topics. The convex geometric perspective of topic models gave rise to some interesting questions. This paper tackles the problem of finding an appropriate number of topics by topic simplex covering algorithm called conic scan coverage approach. Overall, this paper is clearly written, and easy to follow. Theoretical justifications of CSC support the main claim of the paper as well. However, unlike the CSC algorithm, the document CSC algorithm contains some arbitrary steps such as the mean shifting and spherical k-means. The justification of these steps may bring some other questions. For example, is the mean shifting result always around the initial direction? if not, is the initializing direction in step 4 of algorithm2 necessary? A random selection of an initial direction would be enough since it would also cover the topic simplex, eventually. A similar question arises from the spherical k-means; the resulting topics are no longer a vertex of topic convex. What if the variance of the original topic vertex before these steps is relatively low than the others? (may be a long document could not be the topic vertex?) Do we still need the mean-shifting and k-means in this case? or is it better to keep these vertices? It would be more interesting if there are some (empirical?) discussions on the variances of the point estimate and the proposed steps.

Reviewer 3



* Summary This paper introduces a novel algorithm, Conic Scan Coverage, that is based on convex geometry ideas and can perform non-parametric topic modeling. The algorithm is intuitively based on the idea of covering the topic simplex with cones. The papers presents the results of an experimental evaluation and the supplementary contains detailed proofs and example topics inferred by the algorithm. * Evaluation I have very mixed feelings about this paper. On the one hand, I must say I had a lot of fun reading it. I am really fond of the convex geometric approach to topic modeling, it is such an original and potentially promising approach. Indeed, the results are very good, and the topics presented in the supplementary material look great. On the other hand, I have a lot of uncertainty about this paper. First, I must confess I do not understand the paper as well as I would like. Of course, it could mean that I should have tried harder, or maybe that the authors should explain more. Probably a bit of both. Second, the experiments are weak. However, the paper is so original that I don't think it should be rejected just because of this. To summarize: + Impressive results: excellent runtime and statistical performance, good looking topics - Weak experimental evaluation + Very original approach to topic modeling - Algorithm 1 seems very ad-hoc, and justification seems insufficient to me Could you please show me a few example of inferred topic distribution? * Discussion - The experimental evaluation could be much better. The python implementation of Gibbs-LDA is pretty bad, it makes for a very poor baseline. You really need to say somewhere how many iterations were run. When comparing against algorithms like LDA Gibbs or HDP Gibbs, as in table 1, you can't report a single number. You should plot the evolution of perplexity by iteration. For large datasets, Gibbs LDA can reach a good perplexity and coherence in surprisingly few iterations. Also, there should be some error bars, a number in itself is not very convincing. You should also share information about hyperparameter settings. For instance, HDP and LDA can exhibit very different perplexity for different values of their alpha and beta hyperparameters. Finally, you should also vary the vocabulary size. I did note that V is very small, and it is worrysome. There are been several algorithms proposed in the past that seemed very effective for unrealistic small vocabulary sizes and didn't scale well to this parameter. - The algorithm seems very ad-hoc I am surprised that you choose a topic as the furthest document and then remove all the documents within some cosine distance. Why makes such a farthest document a good topic representative? Also, if we remove the documents from the active sets based solely on one topic, are we really trying to explain the documents as mixture of topics? I would be really curious to see a few documents and their inferred topic distributions to see if it is interpretable. - What good is Theorem 4? Theorem 4 is very interesting and definitely provides some good justification for the algorithm. However, it assumes that the number of topics K is known and doesn't say much about why such a procedure should find a good number of topics. Indeed, in the end, I don't have much intuition about what exactly drives the choice of number of topics.